# Liquid Crystal Nanoparticle Conjugates for Scavenging Reactive Oxygen Species in Live Cells

**DOI:** 10.3390/ph15050604

**Published:** 2022-05-14

**Authors:** Okhil K. Nag, Jawad Naciri, Kwahun Lee, Eunkeu Oh, Bethany Almeida, James B. Delehanty

**Affiliations:** 1Center for Bio/Molecular Science and Engineering, Naval Research Laboratory (NRL), Code 6900, 4555 Overlook Ave. SW, Washington, DC 20375, USA; jawad.naciri@nrl.navy.mil (J.N.); kwahun.lee.ctr.ks@nrl.navy.mil (K.L.); james.delehanty@nrl.navy.mil (J.B.D.); 2American Society for Engineering Education, Washington, DC 20036, USA; balmeida@clarkson.edu; 3Naval Research Laboratory, Optical Sciences Division, Code 5600, 4555 Overlook Ave. SW, Washington, DC 20375, USA; eunkeu.oh@nrl.navy.mil; 4Department of Chemical and Biomolecular Engineering, Clarkson University, 8 Clarkson Ave., Potsdam, NY 13699, USA

**Keywords:** liquid crystal nanoparticles, ROS scavenger, oxidative stress, TEMPO, lipid peroxidation, reactive oxygen species

## Abstract

The elevated intracellular production of or extracellular exposure to reactive oxygen species (ROS) causes oxidative stress to cells, resulting in deleterious irreversible biomolecular reactions (e.g., lipid peroxidation) and disease progression. The use of low-molecular weight antioxidants, such as 4-amino-2,2,6,6-tetramethylpiperidine-1-oxyl (TEMPO), as ROS scavengers fails to achieve the desired efficacy because of their poor or uncontrolled cellular uptake and off-target effects, such as dysfunction of essential redox homeostasis. In this study, we fabricated a liquid crystal nanoparticle (LCNP) conjugate system with the fluorescent dye perylene (PY) loaded in the interior and poly (ethylene glycol) (PEG) decorated on the surface along with multiple molecules of TEMPO (PY-LCNP-PEG/TEMPO). PY-LCNP-PEG/TEMPO exhibit enhanced cellular uptake, and efficient ROS-scavenging activity in live cells. On average, the 120 nm diameter PY-LCNPs were conjugated with >1800 molecules of TEMPO moieties on their surface. PY-LCNP-PEG/TEMPO showed significantly greater reduction in ROS activity and lipid peroxidation compared to free TEMPO when the cells were challenged with ROS generating agents, such as hydrogen peroxide (H_2_O_2_). We suggest that this is due to the increased local concentration of TEMPO molecules on the surface of the PY-LCNP-PEG/TEMPO NPs, which efficiently bind to the plasma membrane and enter cells. Overall, these results demonstrate the enhanced capability of TEMPO-conjugated LCNPs to protect live cells from oxidative stress by effectively scavenging ROS and reducing lipid peroxidation.

## 1. Introduction

Reactive oxygen species (ROS) play critical roles in vitro and in vivo in controlling normal cellular functions and the progression of many acute and chronic diseases, including cancer, arthritis, cardiovascular disease, and neurodegenerative disorders [1,2,3]. Low levels of ROS are required to maintain physiological functions, including cellular signaling and proliferation, host defense, and gene expression. However, cells are exposed to damaging oxidative stress when excessive ROS, such as superoxide radical anion (O_2_^•−^), hydroxyl radical (HO^•^), singlet oxygen (^1^O_2_), and hydrogen peroxide (H_2_O_2_), are generated in the cells [2,4,5]. The deleterious effects of ROS occur at the structural (e.g., protein denaturation) and molecular (e.g., lipid, protein, and DNA peroxidation) levels at both the plasma membrane and intracellularly [5,6]. Controlling the ROS level and oxidative stress is essential for maintaining normal cellular function, survivability, and controlling diseases [7]. Thus, ROS-scavenging agents that can effectively neutralize extracellular and intracellular ROS are of particular interest in cell biology and biomedicine to maintain and modulate cellular functions [8].

Exogenous ROS-scavenging agents encompass a class of low-molecular weight molecules that are capable of neutralizing ROS species by various mechanisms [9]. However, ROS scavengers have several inherent limitations, including their short half-life in vivo, cytotoxicity, inadequate or excessive cellular uptake, and off-target effects (e.g., interference with cellular pathways such as the electron transport chain) [10,11,12]. Nitroxides, a class of stable radicals containing a nitroxyl group (N-O^•^), are attractive due to their redox properties that catalytically neutralize ROS (e.g., superoxide ion radicals and their protonated forms, such as hydroperoxyl radicals) in a mechanism similar to superoxide dismutase (SOD), an endogenous ROS scavenger [9,13,14]. Nitroxides are particularly effective in neutralizing ROS that initiate peroxidation of lipids and proteins in cells, and they have been widely studied as ROS scavengers [9,10]. Recently, several studies have reported on 4-amino-2,2,6,6-tetramethylpiperidine-1-oxyl (TEMPO), a nitroxide derivative, and its activity as a ROS-scavenging agent when loaded into the core of self-assembled polymeric nanoparticles (NPs) [15,16,17,18,19]. Yoshitomi et al. reported TEMPO-containing poly (ethylene glycol)-b-poly (methylstyrene) (PEG-b-PMS) block copolymer micelles as pH-activated ROS-scavenging agents for the relief of acute kidney injury [15]. These formulations minimized off-target effects, improved the pharmacokinetic properties of TEMPO, and acted as a ROS scavenger through release or exposure of TEMPO moieties to cells/tissue via NP disassembly in the presence of stimuli such as pH. Although such core-loaded TEMPO-NP systems have proven effective for particular applications (e.g., targeting tumors or specific tissues), the display of TEMPO on the NP surface could be a more effective strategy for the controlled modulation of the ROS-scavenging activity of TEMPO. Multiple molecules of TEMPO conjugated on the surface of NPs can provide increased local concentration and multivalent avidity, thereby increasing its ROS-scavenging activity. There are limited reports of the direct assessment of ROS-scavenging activity of TEMPO on the surface of the NPs [20]. Recently, Li et al. reported TEMPO-conjugated inorganic NPs (AuNP, ~40 nm) to control ROS levels in mesenchymal stem cells (MSCs), which require a distinct level of ROS for differentiation [20]. 

We previously demonstrated the utility of the liquid crystal nanoparticle (LCNP) platform for the targeted delivery of dyes and drug cargos to either the plasma membrane or to the cell interior [21,22,23]. For example, we showed the specific plasma membrane-targeted delivery of the potentiometric dye 3,3′-dioctadecyloxacarbocyanine perchlorate (DiO) by loading DiO into the core and conjugating cholesterol on the surface of the LCNP [22]. This approach localized the delivery of DiO to the membrane while attenuating its cytotoxicity by ~40%. We also showed the ability of the LCNP delivery system to simultaneously serve as both scaffold and energy donor to augment the cell-killing efficiency of the ROS-generating moiety zinc(II) phthalocyanine [23]. In the study detailed herein, we extend the utility of LCNPs to realize a conjugate system for the augmentation of the ROS-scavenging activity of TEMPO. Perylene (PY)-loaded LCNPs (PY-LCNPs) were surface functionalized with poly (ethylene glycol) (PEG) and TEMPO (PY-LCNP-PEG/TEMPO) via 1-ethyl-3-(3-dimethylaminopropyl)carbodiimide (EDC) conjugation chemistry. The ROS-scavenging activity of PY-LCNP-PEG/TEMPO was quantified in live cells challenged with intracellular and extracellular ROS-producing agents, such as H_2_O_2_ and tert-butyl hydroperoxide (tBHP), using confocal imaging of a fluorescent ROS probe in live cells. In human cervical adenocarcinoma (HeLa) cells exposed to extracellular ROS-producing H_2_O_2_, the PY-LCNP-PEG/TEMPO showed >300% enhancement of ROS scavenging activity and a 26% reduction in lipid peroxidation compared to cells incubated with free TEMPO (TEMPO_free_) in bulk solution. We attribute this effect to the locally concentrated display of TEMPO on the surface of the PY-LCNP coupled with the efficient cellular binding and internalization of PY-LCNP-PEG/TEMPO. Overall, these results demonstrate the enhanced capability of TEMPO-conjugated LCNPs to protect live cells from oxidative stress by effectively scavenging ROS and reducing lipid peroxidation.

## 2. Results and Discussion

### 2.1. Rationale for the Synthesis of PY-LCNP-PEG/TEMPO

NPs have been studied extensively for their ability to deliver various cargos (drugs, dyes, proteins) to cells both in vitro and in vivo [24,25,26]. One of the primary advantages of using NPs as a delivery scaffold is the incorporation of multiple copies of cargo on the NP surface for enhanced activity through multivalent avidity and increased local concentration. The goal of this study was to ascertain whether, through conjugation and multivalent display of TEMPO on the surface of dye (PY)-loaded LCNPs, the ROS-scavenging activity of TEMPO could be augmented compared to TEMPO_free_ in bulk solution. We synthesized and characterized fluorescently trackable LCNPs that were loaded with PY in their hydrophobic core and covalently conjugated to poly(ethylene glycol) (PEG) and the ROS-scavenging moiety TEMPO (Figure 1A) on the surface for improved colloidal stability and efficient ROS-scavenging capabilities, respectively.

### 2.2. Synthesis and Characterization of PY-LCNP-PEG/TEMPO

PY-loaded LCNPs (PY-LCNP) were synthesized following the basic procedures described previously [21,22]. The as-synthesized PY-LCNPs display negatively charged carboxylate groups on their surface, providing colloidal stability in aqueous media and presenting a chemical handle for covalent conjugation to amine-containing functional molecules, such as TEMPO. We performed EDC coupling to the PY-LCNPs using a mixed ratio of PEG_2000_-NH_2_ (30 mol%, 300 µM) and TEMPO (70 mol%, 700 µM) (Figure 1A) [23]. After conjugation, the particles were purified and characterized by measuring their electrophoretic mobility, size, charge, and spectroscopic properties. The PEG/TEMPO-conjugated PY-LCNP (PY-LCNP-PEG/TEMPO) showed meaningfully different electrophoretic mobility compared to the unconjugated PY-LCNPs, indicating successful conjugation of PEG/TEMPO onto the PY-LCNP surface (Figure 1B). As anticipated, negatively charged, unconjugated PY-LCNP showed clear and robust mobility toward the cathode. However, the PY-LCNP-PEG/TEMPO exhibited minimal migration toward the cathode, confirming the decrease in negative surface charges as a result of the conjugation. These data provided strong evidence for the successful conjugation of PEG/TEMPO onto the PY-LCNP surface. We used dynamic light scattering to estimate the hydrodynamic size and particle concentration of PY-LCNP-PEG/TEMPO. We observed a significant increase (~24 nm) in the average hydrodynamic diameter (Figure 1C) from 121 ± 5 nm (polydispersity index (PDI) = 0.07) for PY-LCNP to 145 ± 14 nm (PDI = 0.19) for PY-LCNP-PEG/TEMPO. The estimated particle concentration for PY-LCNP-PEG/TEMPO was ~53 nM. Zeta potential measurements confirmed the change in overall particle charge from −37 ± 1 mV (PY-LCNP) to −1 ± 0.7 mV (PY-LCNP-PEG/TEMPO) (Figure 1C), which is consistent with the results observed during gel electrophoresis. The conjugation of TEMPO to the PY-LCNP was further characterized and quantified by UV-Vis spectroscopy. The spectra in Figure 1C show distinct differences between the PY-LCNPs and PY-LCNP-PEG/TEMPO in the spectral region from 450 nm and below due to the presence of TEMPO and PEG on the NPs. The extent of conjugation of TEMPO onto the PY-LCNP surface was determined spectroscopically using the TEMPO absorbance maximum at 426 nm (data not shown) and extinction coefficient (17.34 M^−1^ cm^−1^) [27]. TEMPO absorbance on the LCNP was deconvoluted by subtracting the absorbance of the LCNP. The calculated concentration of TEMPO on the PY-LCNP-PEG/TEMPO was determined to be ~100 µM in a solution that was 53 nM in NP concentration, indicating ~1880 TEMPO molecules conjugated to each PY-LCNP. According to the initial reaction conditions in the EDC/NHS reaction, this corresponds to an efficiency of ~14% for the conjugation of TEMPO to the PY-LCNP. Additionally, the fluorescence (FL) emission spectra of PY in the PY-LCNP-PEG/TEMPO was unchanged from that for PY in the unconjugated LCNPs, indicating no change in PY spectral properties as a result of conjugation. Fully characterized particles were stored at 4 °C and used within 3 weeks, with no visually detectable precipitation or aggregation. This is likely attributable to the presence of PEG, which is well known for imparting colloidal and steric stability to NPs, on the surface of the particles.

### 2.3. Cellular Uptake of PY-LCNP-PEG/TEMPO

Next, we studied the interaction of the PY-LCNP-PEG/TEMPO with live HeLa cells in a time-dependent manner using confocal microscopy. The PY-LCNP-PEG/TEMPO molecules were incubated with cells at 37 °C for 100 min, and images were acquired every 10 min using the red channel for FL emission signal collection (Figure 2A). Upon 10 min of incubation, PY-LCNP-PEG/TEMPO clearly started accumulating at the plasma membrane, the extent of which increased and became more uniform with increasing incubation time. Quantification of FL emission intensity revealed that the degree of membrane accumulation of the conjugates reached its maximum at ~40 min (Figure 2B), and the staining pattern remained largely uniform, with only a few discrete punctate regions. At the 2 h timepoint, however, it was apparent that the PY-LCNP-PEG/TEMPO began to be internalized by the cells. To determine the rate of PY-LCNP-PEG/TEMPO cellular internalization, we performed time-resolved colocalization studies using an AlexaFluor^TM^ 488 conjugate of the lectin, wheat germ agglutinin (WGA-AF488), which binds to *N*-acetyl-d-glucosamine and sialic acid moieties on the plasma membrane.

The cells were incubated with PY-LCNP-PEG/TEMPO for 30 min, washed, and then at 1 h intervals, the plasma membrane was labeled with WGA-AF488 and imaged. Figure 2C shows merged images (DIC, red, and green channels) taken over a 6 h time period. It is important to note here that due to the broad emission of the PY dye when excited at 488 nm, there is a signal from the PY-LCNP-PEG/TEMPO in both the green and red channels, while the WGA-AF488 probe emits a signal in the green channel only. Thus, we expect ~100% colocalization of red to green because green pixels will always overlap red pixels as a result of this overlapping signal from the PY-LCNP-PEG/TEMPO. However, we expect to see changes in green to red colocalization as more and more PY-LCNP-PEG/TEMPO is internalized and no longer overlaps the green pixels. Indeed, the data showed that at the 1 h timepoint, ~87% of the PY-LCNP-PEG/TEMPO was colocalized with the green WGA-AF488 membrane marker (as determined by Pearson’s colocalization coefficient (PCC)), suggesting that most of the PY-LCNP-PEG/TEMPO was localized to the plasma membrane. By 2 h, the percentage of green WGA-AF488 signal on the membrane increased and the colocalization of green to red decreased to ~60%, indicating that the PY-LCNP-PEG/TEMPO molecules were steadily internalized by the cells while the WGA-AF488 marker remained resident on the membrane (see ‘green to red’ plot in Figure 2D). This colocalization reached its maximum at ~3 h (PCC = ~50%). The colocalization of red to green (used here to track PY-LCNP-PEG/TEMPO molecules themselves) remained unchanged over the entire 6 h experimental window as expected (see ‘red to green’ plot in Figure 2D). The PY-LCNP-PEG/TEMPO adopted a punctate, intracellular morphology over time, suggestive of encapsulation of PY-LCNP-PEG/TEMPO within endocytic vesicles. These observations of PY-LCNP-PEG/TEMPO cellular internalization are in contrast to our previous studies where LCNPs (conjugated to PEG-cholesterol moieties) remained resident on the plasma membrane for up to 4 h, where they were associated with lipid-raft domains and enabled the specific delivery of the potentiometric imaging dye, DiO [22]. Additionally, in this study, we could not directly compare the cellular uptake of PY-LCNP-PEG/TEMPO to that of TEMPO_free_ due to the lack of inherent fluorescence of the latter. Previous studies of the cellular delivery of TEMPO_free_ from bulk solution showed minimal cellular internalization and inability to cross the plasma membrane bilayer (presumably due to its charged nature from the primary amine), resulting in limited ROS scavenging efficacy of TEMPO_free_ [14]. Thus, based on this prior literature, this suggests that our PY-LCNP-PEG/TEMPO would improve cellular uptake of TEMPO due to the overall modulation of its properties (e.g., non-charge nature due to the lack of primary amine after conjugation) attributed by the LCNP.

### 2.4. ROS-Scavenging and Cellular Protection Efficiency of PY-LCNP-PEG/TEMPO

We sought to study the ROS-scavenging and cellular protection activity of the PY-LCNP-PEG/TEMPO in live HeLa cells challenged with two different ROS-generating agents: H_2_O_2_ (as an extracellular ROS source) and tBHP (as an intracellular ROS generator). H_2_O_2_ is a low-reactive ROS molecule, but it can generate highly reactive forms of ROS, such as hydroxyl radicals (OH^•^), in cellular environments [28,29]. In contrast, tBHP generates a higher degree of ROS, including H_2_O_2_, alkoxyl, and peroxyl radicals, in the cell through metabolism after cellular internalization [30,31]. Both H_2_O_2_ and tBHP induce significant oxidative stress to cells through multiple mechanisms, including lipid peroxidation, DNA damage, and depletion of cellular glutathione levels [29]. To perform these experiments, HeLa cells were first incubated with equivalent concentrations of TEMPO (50 µM), present as either TEMPO_free_ or PY-LCNP-PEG/TEMPO (27 nM of LCNP), followed by exposure to the ROS-generating agents. ROS activity in the cells was imaged and quantified every 10 min for 60 min by confocal microscopy using the fluorescent probe H2DCFDA, which positively tracks the presence of ROS in cells as a function of increasing fluorescence emission. We first tested the ROS-scavenging and cellular protection activity against H_2_O_2_ (500 µM) exposure. As shown in Figure 3A, minimal fluorescence emission for H2DCFDA was observed immediately after adding the ROS-generating agent H_2_O_2_ (Figure 3A, t = 0). However, after 60 min of H_2_O_2_ (500 µM) exposure, the emission intensity for the H2DCFDA showed differential increases in untreated, TEMPO_free_, and PY-LCNP-PEG/TEMPO-treated cells. This was also coupled with various levels of membrane blebbing, which is an indication of cellular damage due to ROS activity [32,33]. After 60 min, the intensity of the H2DCFDA in the untreated (control) cells was significantly higher than that of cells treated with PY-LCNP-PEG/TEMPO. Furthermore, there were differences noted in the H2DCFDA signal depending on whether the cells were pretreated with PY-LCNP-PEG/TEMPO for 3 h or immediately prior to H_2_O_2_ exposure (Figure 3A). In this instance, treating the cells with PY-LCNP-PEG/TEMPO immediately prior to H_2_O_2_ exposure proved to be more effective at reducing overall ROS generation and providing protection against cellular damage. Additionally, the DIC imaging of immediately treated PY-LCNP-PEG/TEMPO cells showed noticeably lower plasma membrane blebbing compared to the untreated or TEMPO_free_ treated cells. Given our results for the cellular internalization of PY-LCNP-PEG/TEMPO (see Figure 2 above), it is likely that the large degree of cellular uptake of PY-LCNP-PEG/TEMPO over 3 h depletes the local TEMPO concentration present at the plasma membrane, resulting in reduced efficacy in the prevention of H_2_O_2_-induced ROS generation. To quantify and compare the ROS activity and rate of ROS neutralization in different samples, we analyzed the cells for the changes in H2DCFDA FL emission intensities every 10 min over 60 min of H_2_O_2_ exposure. As shown in Figure 3B, ROS activity in the untreated cells gradually increased with incubation time (500 µM H_2_O_2_) and reached an activity level 34-fold above that of the control cells (rate of increase = 52% min^−1^). In comparison, the ROS activity in the cells treated with TEMPO_free_ and PY-LCNP-PEG/TEMPO immediately prior to H_2_O_2_ treatment increased only 12-fold (rate of increase = 19% min^−1^) and 4-fold (rate of increase = 4% min^−1^), respectively. However, the ROS activity of cells treated with PY-LCNP-PEG/TEMPO and incubated for 3 h prior to H_2_O_2_ treatment increased 9-fold (rate of increase = 12% min^−1^), which is more than 2-fold higher and faster than the cells exposed to H_2_O_2_ immediately after the PY-LCNP-PEG/TEMPO treatment. These results strongly suggest that, compared with TEMPO_free_, PY-LCNP-PEG/TEMPO pre-treatment immediately prior to H_2_O_2_ exposure reduced the overall ROS activity by 3-fold and significantly decreased the rate of cellular ROS exposure and increased protection against H_2_O_2_-incuded ROS activity. Meanwhile, pre-treatment for 3 h of PY-LCNP-PEG/TEMPO reduced overall ROS activity by ~1.3-fold compared with TEMPO_free_. In the context of previous reports, our PY-LCNP-PEG/TEMPO conjugate system is an improvement over other NP-TEMPO systems. For example, Li et al. reported on gold NP (AuNP)-TEMPO conjugates, wherein the ROS scavenging activity of TEMPO was improved by ~1.3-fold when displayed on the surface of 40 nm diameter AuNPs [20]. The enhanced activity we observed in this study is likely due to the greater degree of multivalent display in our PY-LCNP-PEG/TEMPO system (>1800 TEMPO moieties per NP) on the 120 nm diameter LCNPs.

To further characterize the ROS-scavenging and cellular protection capabilities of PY-LCNP-PEG/TEMPO, we tested the conjugate’s ability to protect cells against tBHP, which generates ROS species intracellularly upon its internalization and metabolism [30,31]. The ROS-scavenging activity of PY-LCNP-PEG/TEMPO (50 µM TEMPO or 27 nM LCNP) was tested against two tBHP concentrations: 100 µM and 500 µM. In this case, we anticipated that internalized PY-LCNP-PEG/TEMPO might show better ROS-scavenging activity against tBHP-induced ROS species compared with membrane-associated PY-LCNP-PEG/TEMPO as TEMPO has previously been shown to have a low capacity for traversing the plasma membrane [14]. As shown in Figure 4A, tBHP generated an increasing concentration of ROS species over time in untreated cells, reaching up to 32-fold (rate of increase = 80% min^−1^) and 50-fold (rate of increase = 100% min^−1^) for 100 µM and 500 µM of tBHP, respectively, over 60 min. PY-LCNP-PEG/TEMPO showed robust ROS scavenging capability against tBHP-induced ROS species when tested immediately after incubation. With immediate PY-LCNP-PEG/TEMPO treatment, the ROS activities against 100 µM and 500 µM tBHP were measured to be 3.6-fold (rate of increase = 0.2% min^−1^) and 11-fold (rate of increase = 6.4% min^−1^) lower, respectively, compared to untreated cells. However, lower ROS-scavenging ability of PY-LCNP-PEG/TEMPO was measured when cell samples were exposed to tBHP 3 h after labeling with PY-LCNP-PEG/TEMPO, contrary to our initial thoughts. Here, ROS production against 100 µM and 500 µM tBHP was measured to be about 4-fold (rate of increase = 14% min^−1^) and 3-fold (rate of increase = 31% min^−1^) lower, respectively, compared to untreated cells. This lower ROS-scavenging ability of PY-LCNP-PEG/TEMPO upon internalization is likely due to limited endocytic escape and the clustering behavior of PY-LCNP-PEG/TEMPO observed inside cells (see Figure 2C). Similar trends were also observed (Figure 4B) when comparing the ROS-scavenging activity of PY-LCNP-PEG/TEMPO against the same concentration of H_2_O_2_ and tBHP (500 µM). We observed 8-fold and 11-fold reductions in ROS activity against H_2_O_2_ and tBHP-induced ROS activity, respectively, when bound to the plasma membrane. After internalization, PY-LCNP-PEG/TEMPO demonstrated only 4-fold and 3-fold reductions in ROS activity against H_2_O_2_ and tBHP-induced ROS activity, respectively.

Next, we sought to evaluate the protective effects of TEMPO against oxidative lipid peroxidation induced by H_2_O_2_ and tBHP. To do this, we measured the levels of 4-hydroxynonenal (4-HNE), an unsaturated hydroxyalkenal that is produced by lipid peroxidation, including the peroxidation of membrane phospholipids [34]. Here, we assayed for 4-HNE in the 1–2 h window after H_2_O_2_ and tBHP exposure, where cellular uptake data demonstrated that the PY-LCNP-PEG/TEMPO conjugate remained largely resident on the plasma membrane. In cells not treated with TEMPO_free_ or PY-LCNP-PEG/TEMPO, exposure to H_2_O_2_ or tBHP resulted in increases of 45% and 85%, respectively, in 4-HNE levels (Figure 5A,B). In the presence of TEMPO_free_ or PY-LCNP-PEG/TEMPO, this response was attenuated. In the case of H_2_O_2_, increases of 34% and 8% in 4-HNE levels were observed when the cells were labeled with TEMPO_free_ and PY-LCNP-PEG/TEMPO, respectively. For tBHP, 4-HNE levels of 57% and 25% compared to control were noted for TEMPO_free_ and PY-LCNP-PEG/TEMPO, respectively. These results demonstrate that TEMPO protects cells from lipid peroxidation, and the degree of lipid peroxidation inhibition is 26% (against H_2_O_2_ exposure) and 32% (against tBHP exposure) higher in cells treated with PY-LCNP-PEG/TEMPO versus those treated with TEMPO_free_. No significant difference was observed when the lipid peroxidation inhibition levels were compared between H_2_O_2_ and tBHP. This is consistent with reports detailing that both ROS-generating agents induce similar levels of lipid peroxidation, particularly in the plasma membrane [35,36]. Additionally, NP-based ROS-scavenging agents, such as nanoceria, have been reported to inhibit lipid peroxidation at the mitochondrial level and deter the progression of Parkinson’s disease [31]. One of the implications of our PY-LCNP-PEG/TEMPO can be the inhibition of lipid peroxidation in subcellular locations, such as mitochondria, when designed to be specifically delivered.

Given the efficient ROS-scavenging and cellular protection efficacy against ROS of PY-LCNP-PEG/TEMPO, we sought to compare the cellular viability of the TEMPO_free_ and PY-LCNP-PEG/TEMPO on live cells. As shown in Figure 6A,B, cellular viability decreases with increasing concentration of TEMPO in both its free and conjugated forms. At ~50 µM TEMPO in free or conjugated form, cellular viability is calculated to be ~85% and 82%, respectively; we chose this concentration (50 µM of TEMPO, equivalent) for the cell delivery/labeling and ROS-scavenging experiments. These results suggest that PY-LCNP-PEG/TEMPO exhibits comparable cellular viability to TEMPO_free_. Of critical importance was the observed ~85% cell viability for the full ensemble PY-LCNP-PEG/TEMPO (at 50 µM TEMPO concentration), which confirmed the high degree of biocompatibility of the full conjugate system.

## 3. Materials and Methods

### 3.1. Materials

All chemicals, including 4-amino-2,2,6,6-tetramethylpiperidine-1-oxyl (TEMPO, CAS# 14691-88-4), were purchased from Millipore Sigma (St. Louis, MO, USA) and used as received unless otherwise mentioned. The perylene (PY) dye was prepared as previously described [37]. Dulbecco’s phosphate-buffered saline (DPBS), 4-(2-hydroxyethyl)-1-piperazineethanesulfonic acid (HEPES, 1M), Dulbecco’s Modified Eagle Medium (DMEM) containing 25 mM HEPES (DMEM-HEPES), live cell imaging solution (LCIS), 1-ethyl-3-(3-dimethylaminopropyl)carbodiimide hydrochloride (EDC·HCl, CAS# 25952-53-8), N-hydroxysulfosuccinimide sodium salt (NHS, CAS# 106627-54-7), and the plasma membrane probe wheat germ agglutinin-Alexa Fluor™ 488 conjugate (WGA-AF488) were purchased from Thermofisher Scientific (Waltham, MA, USA). Poly (ethylene glycol) amine hydrochloride (PEG-NH_2_.HCl, MW 2000) was purchased from Nanocs Inc. (New York, NY, USA). Human 4-HNE competitive ELISA 96-well assay kit was purchased from LifeSpan Biosciences, Inc. (Seattle, WA, USA). UV-Vis absorbance and fluorescence spectra of the LCNPs were measured on a UV-1900i Spectrophotometer (Shimadzu Scientific Instruments, Columbia, MD, USA) and an RF-6000 Spectrofluorophotometer (Shimadzu Scientific Instruments, Columbia, MD, USA), respectively.

### 3.2. Synthesis of PY-LCNP-PEG/TEMPO Nanoparticle Conjugates

First, bare PY-LCNPs were synthesized using a two-phase miniemulsion procedure as described previously [21,22,37,38]. Briefly, liquid crystalline diacrylate cross-linking agent (DACTP11; 45 mg), perylene derivative (PY; 2 mg), and a free radical initiator (azobisisobutyronitrile, 1 mg) for polymerization were dissolved in 2 mL chloroform and added to an aqueous solution of acrylate-functionalized surfactant (AC10COONa, 13 mg in 7 mL). The mixture was then heated to 64 °C under an N_2_ atmosphere to initiate the polymerization of both the cross-linking agent and surfactant as the chloroform evaporated, leaving the PY-LCNP suspension stabilized by the surfactant. Following synthesis, the NP suspension was filtered (3×) through a 0.2 µm syringe filter to reduce the average particle size and polydispersity, and any aggregated NPs that appeared in the pellet were discarded. To synthesize PY-LCNP-PEG/TEMPO particles, the PY-LCNP were covalently conjugated with a mixture of TEMPO (700 μM) and PEG_2000_-NH_2_.HCl (300 μM) (mole ratio 7:3) via EDC/NHS chemistry as described previously [22]. After the conjugation reaction, particles were purified using a PD10 column (0.1× DPBS as eluent) and characterized for their physicochemical properties, as described below.

### 3.3. Physicochemical Properties of PY-LCNP-PEG/TEMPO

Conjugation of TEMPO and PEG_2000_-NH_2_.HCl to the carboxylate group on the surface of the PY-LCNPs was confirmed by agarose (1%) gel electrophoresis under an applied voltage of 95 V for 30 min at room temperature in 1× tris-buffered saline (TBS). Particle size distribution and sample concentration were measured by dynamic light scattering (DLS) of LCNP solution (40× dilution of the original suspension) in water using a ZetaSizer NanoSeries equipped with a HeNe laser source (λ = 633 nm) (Malvern Instruments Ltd., Worcestershire, UK) and analyzed using Dispersion Technology Software (DTS, Malvern Instruments Ltd., Worcestershire, UK). Zeta-potential was measured on a ZetaSizer NanoSeries equipped with a HeNe laser source (λ = 633 nm) (Malvern Instruments Ltd., Worcestershire, UK) and an avalanche photodiode for detection. For each analysis, at least four measurements were performed, and the data were reported as average values ± standard error of the mean (SEM).

### 3.4. Interaction of PY-LCNP-PEG/TEMPO Conjugates with HeLa Cells

Cellular labeling efficiency and uptake of PY-LCNP-PEG/TEMPO were quantified by measuring fluorescent emission intensities in real time using HeLa cells (ATCC© CCL-2TM, Manassas, VA, USA) between passages 5 and 15. The cells were seeded on a 35 mm petri dish with 14 mm glass-bottom insert (#1.5 cover glass, MatTek Corp., Ashland, MA, USA) at a density of ~7 × 10^4^ cells/mL (3 mL/well), and were cultured for 24 h in standard incubation conditions before incubating with PY-LCNP-PEG/TEMPO. Solutions of PY-LCNP-PEG/TEMPO (27 nM in LCIS) were added directly to the cell monolayers in a microscope stagetop imaging incubation chamber at 37 °C. For experiments to determine uptake of the PY-LCNP-PEG/TEMPO alone (no membrane labeling dye), the cells were imaged (60× oil immersion objective) every 10 min for 2 h by DIC and confocal laser scanning microscopy (CLSM) using 488 nm HeNe laser excitation with 570–620 nm (red) fluorescence detection filter using a Nikon A1RSi confocal microscope (Nikon Co. Ltd., Tokyo, Japan). Cellular labeling efficiency was calculated by measuring fluorescent emission intensities on the cells by drawing the region of interest (ROI) at different time points and represented as normalized intensities compared to 10 min image. For internalization kinetics, the cells were first labeled with PY-LCNP-PEG/TEMPO by incubating for 30 min, followed by washing and incubating up to 6 h while imaging every 1 h after labeling with the plasma membrane probe WGA-AF488. Internalization of the particles was imaged with CLSM using 488 nm HeNe laser excitation with 500-550 nm (green) and 570–620 nm (red) fluorescence detection filters, respectively, using approximate equal gain settings for both the green and red channels. Internalization kinetics of PY-LCNP-PEG/TEMPO were calculated by PCC using NIS-Elements AR 4.3 Software of Nikon A1RSi confocal microscope.

### 3.5. Cellular ROS Scavenging and Protection of PY-LCNP-PEG/TEMPO

The ability of PY-LCNP-PEG/TEMPO and TEMPO_free_ to provide protection against ROS was assayed in HeLa cells. The cells were seeded on 35 mm petri dishes with 14 mm glass-bottom insert (#1.0 cover glass, MatTek Corp., Ashland, MA, USA) and cultured for 24–48 h to a 70–80% confluence. The cell monolayers were incubated with PY-LCNP-PEG/TEMPO (27 nM particle, 50 µM TEMPO) or TEMPO_free_ (50µM) at 37 °C for 20 min and washed (3×) with DPBS to remove unbound NPs or TEMPO_free_. To measure cellular ROS activity, the cells were further labeled with a fluorescence (green emitting) ROS probe 2′7′-Dichlorodihydrofluorescein diacetate (H2DCFDA) (10 uM) at 37 °C for 15 min followed by washing (3×) with DPBS. The cell samples labeled with the LCNPs and H2DCFDA were then placed in a live cell incubation (37 °C) chamber on an A1RSi confocal microscope and treated with H_2_O_2_ (500 µM, in LCIS) or tert-butyl hydroperoxide (tBHP) (100 µM or 500 µM, in LCIS) for extracellular and intracellular ROS generation, respectively. During incubation, the cells were imaged (60× oil immersion objective) every 10 min for 1 h by DIC and confocal laser scanning microscopy using 488 nm argon laser excitation with a 500–550 nm (green) fluorescence detection filter using a Nikon A1RSi confocal microscope. Image analysis was performed by drawing the ROI in the fluorescent signal on the cells at different time points, representing data as a percent of fluorescent intensity change compared to 0 min.

### 3.6. Lipid Peroxidation Assay

Lipid peroxidation, an indicator of oxidative damage, was evaluated by measuring 4-hydroxy-2-nonenal (4-HNE) levels in cells exposed to H_2_O_2_ in the presence and absence of PY-LCNP-PEG/TEMPO or TEMPO_free_. A human 4-HNE competitive ELISA 96-well assay kit (LifeSpan Biosciences, Inc., Seattle, WA, USA, Cat# LS-F40036-1) was used according to the manufacturer’s protocol. Briefly, HeLa cells were seeded to 96-well tissue culture plates (~5 × 10^3^ cells/well in 100 µL media) and culture to 70–80% confluency. Next, the cells were treated with LCNP-PEG/TEMPO or TEMPO_free_ as described above, washed, and incubated with 500 µM of H_2_O_2_ in DMEM-HEPES for 1 h at 37 °C. The attached cells were washed (3×) with DPBS and trypsinized to collect the cells suspension. Trypsin was removed by centrifugation and the cells were washed (3×) with DPBS and resuspended into DBPS. The cells were lysed by ultrasonication and repeated freeze/thaw cycles and centrifuged (1500× *g* for 10 min) to remove the water-insoluble cellular debris. The presence of 4-HNE in the clear cell lysis were assayed according to the manufacturer’s protocol, and plotted as percent compared to the control cells (not treated with H_2_O_2_). The data were statistically analyzed by the univariate analysis of variance using GraphPad Prism software for Windows (La Jolla, CA, USA). For multiple comparisons, Bonferroni’s post hoc test was applied. All average values are given ± standard error of mean. The acceptable probability for significance was *p* < 0.05.

### 3.7. Cell Viability Assay

Cytotoxicity of the PY-LCNP-PEG/TEMPO and TEMPO_free_ was determined using the CellTiter 96^®^ Aqueous One Solution MTS Cell Proliferation Assay (Promega, Madison WI, USA). Briefly, HeLa cells were seeded to 96-well tissue culture plates (~5 × 10^3^ cells/well in 100 µL media) and culture for 24 h in standard incubation conditions. Next, 50 µL of DMEM-HEPES containing increasing concentrations of PY-LCNP-PEG/TEMPO or TEMPO_free_ were added to the wells and incubated for 30 min at 37 °C. After the incubation, the materials were replaced with 100 µL of complete growth medium and the cells were cultured for 72 h. After this proliferation period, 20 µL of the tetrazolium substrate was added to each well and incubated at 37 °C for 3 h for the formation of color product formazan. The absorbance of the formazan product was read at 590 nm and 650 nm (for subtraction of nonspecific background absorbance) using a Tecan Infinite M1000 (Tecan, Morrisville, NC, USA) microtiter plate reader. Absorbance values with the background subtracted were plotted as a function of material concentration and reported as percent of control cell proliferation (cells cultured in complete media only).

## 4. Conclusions

The advantageous antioxidant properties of TEMPO could find utility in cell biology and biomedicine in protecting cells/tissues from excess ROS activity and oxidative stress when its activity can be controllably modulated in cellular environments. In this study, we utilized fluorescently trackable PY-LCNP as a platform for the multivalent display and delivery of TEMPO to live cells. In summary, we synthesized a TEMPO/PEG-conjugated PY-LCNP with an average diameter of 145 nm wherein each PY-LCNP displayed ~1800 molecules of TEMPO. PY-LCNP-PEG/TEMPO showed significant cellular uptake and enhanced efficacy in reducing ROS activity and lipid peroxidation when the cells were challenged with the ROS-generating agents H_2_O_2_ and tBHP. PY-LCNP-PEG/TEMPO was found to be more effective in scavenging ROS and inhibiting lipid peroxidation when it remained associated with the plasma membrane compared to when it was internalized. This is probably due to the abundant unsaturated lipids in the plasma membrane, which are susceptible to oxidative damage. We expect the PY-LCNP-PEG/TEMPO system described herein will find utility in the protection of live cells against the damaging effects of ROS by minimizing plasma membrane damage and acting as a barrier to shield organelles from ROS invasion and oxidative stress.

## Figures and Tables

**Figure 1 pharmaceuticals-15-00604-f001:**
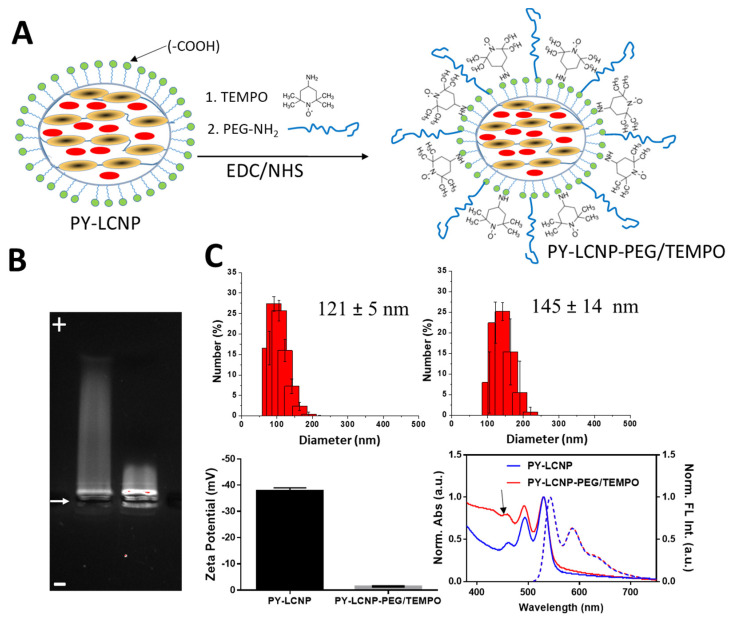
**Synthesis and physicochemical characterization of PY-LCNP-PEG/TEMPO**. (**A**) Schematic of the perylene (PY, red oval)-loaded liquid crystal nanoparticle (PY-LCNP) and EDC conjugation to TEMPO and PEG_2000_-NH_2_. Yellow ovals represent the liquid crystal crosslinker, DACTP11. (**B**) Image of the gel electrophoresis (1% agarose) of the PY-LCNPs. The arrow indicates the location of the sample-loading wells. Unconjugated PY-LCNP (left) migrate further towards the cathode (+) compared to the PEG/TEMPO-conjugated PY-LCNP (PY-LCNP-PEG/TEMPO). (**C**) Distribution plots depicting hydrodynamic size via DLS for PY-LCNPs (top left) and PY-LCNP-PEG/TEMPO (top right), with average diameter (± standard error of the mean (SEM)), zeta potential (bottom left), and UV-Vis absorbance (solid lines) and fluorescence emission (dashed lines) spectra of the PY-LCNPs and PY-LCNP-PEG/TEMPO (bottom right). The arrow indicates the characteristic absorption peak of TEMPO at ~426 nm. The fluorescence emission spectra were collected by excitation at 488 nm.

**Figure 2 pharmaceuticals-15-00604-f002:**
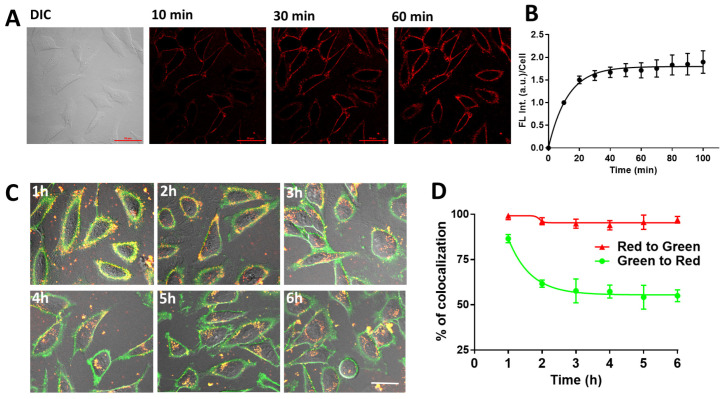
**Cellular uptake of PY-LCNP-PEG/TEMPO**. (**A**) Differential interference contrast (DIC) and confocal fluorescence images of live HeLa cells at 10, 30, and 60 min of incubation with PY-LCNP-PEG/TEMPO conjugates. The red (excitation, 488 nm; emission, 570–620 nm) emission shows PY-LCNP-PEG/TEMPO association with the plasma membrane over time (scale bar, 50 µm). (**B**) Time-resolved quantification of membrane binding of PY-LCNP-PEG/TEMPO during 60 min of incubation. The data represent the emission intensity (avg ± SEM) of 60–80 cells from two independent experiments, normalized to the emission intensity at the beginning of imaging. (**C**) Time-resolved DIC and confocal fluorescence images (merged) of live HeLa cells incubated with PY-LCNP-PEG/TEMPO and labeled with the plasma membrane probe WGA-AF488 (scale bar, 50 µm). (**D**) Time-resolved quantification of the percent of membrane-bound WGA-AF488 (green) and PY-LCNP-PEG/TEMPO (red) signals as a function of incubation time. The data were obtained from Pearson’s colocalization coefficient (PCC, *n* = 3 ± SD) of the green (WGA-AF 488) and green + red (PY) channels, and are expressed as a percentage after normalization to the PCC corresponding to 1 h incubation.

**Figure 3 pharmaceuticals-15-00604-f003:**
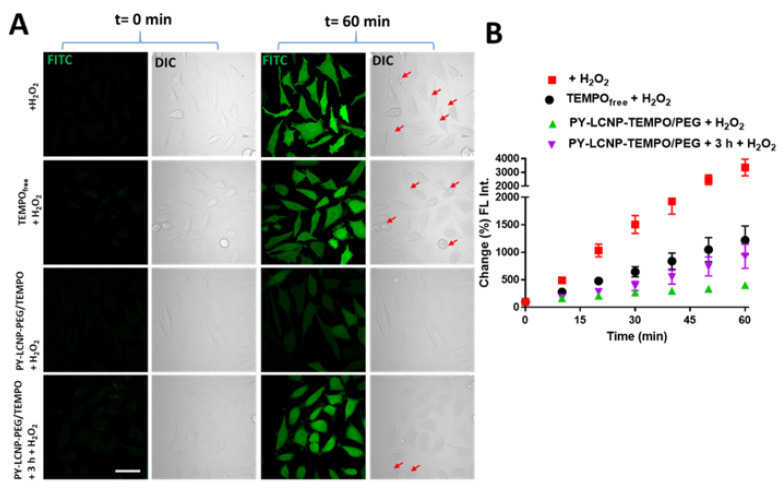
**Quantification of extracellular ROS-scavenging efficiency of PY-LCNP-PEG/TEMPO and TEMPO_free_.** (**A**) DIC and confocal fluorescence images of live HeLa cells at 0 and 60 min of incubation with 500 µM of H_2_O_2_. The fluorescence images show green emission of the ROS probe H2DCFDA in untreated cells, cells treated with TEMPO_free_, and cells treated with PY-LCNP-PEG/TEMPO immediately prior to or 3 h before exposure to H_2_O_2_. The red arrows in the DIC images indicate areas of membrane blebbing (scale bar, 50 µm). (**B**) Time-resolved quantification of the emission intensity of H2DCFDA showing changes (%) in the level of H_2_O_2_-induced ROS in the cells during 60 min incubation. The data for each sample were derived from the green emission intensity of the individual cells (50–80 cells) from three independent experiments (*n* = 3 ± SEM), and are normalized to the emission intensity at the beginning of the imaging.

**Figure 4 pharmaceuticals-15-00604-f004:**
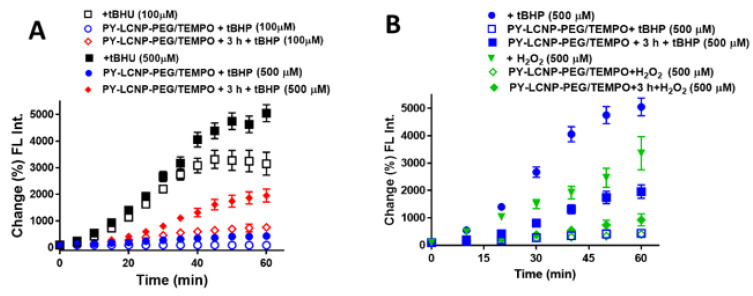
**Quantification of the intracellular ROS-scavenging efficiency of PY-LCNP-PEG/TEMPO**_. (_**A**) Time-resolved quantification of H2DCFDA, which measures ROS activity and shows the change (%) of ROS level in the cells induced by incubation with 100 µM and 500 µM of tBHP with or without PY-LCNP-PEG/TEMPO pre-labeling immediately or after a 3 h incubation. The data for each sample were derived from the emission intensity of the individual cells (50–80 of cells) from three independent experiments (*n* = 3 ± SEM) and are normalized to the emission intensity at the beginning of the imaging. (**B**) A comparative quantification of ROS level in the cells induced with 500 µM of tBHP and H_2_O_2_ with or without PY-LCNP-PEG/TEMPO pre-labeling immediately or after 3 h incubation.

**Figure 5 pharmaceuticals-15-00604-f005:**
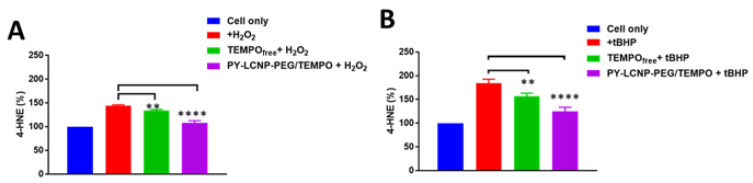
**Lipid peroxidation assay.** Quantification of cellular lipid peroxidation (% of 4-HNE) induced by 500 µM of H_2_O_2_ (**A**) and tBHP (**B**) with or without labeling with PY-LCNP-PEG/TEMPO and TEMPO_free_. Data plotted as change (%) in average (n = 3 ± SEM) compared to control cells (not treated with H_2_O_2_) (** *p* < 0.005; **** *p* < 0.0001).

**Figure 6 pharmaceuticals-15-00604-f006:**
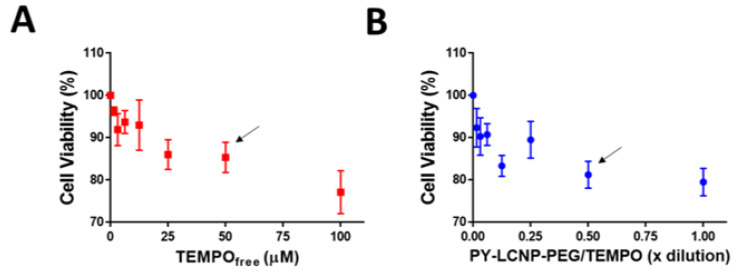
**Quantification of cell viability.** Indicated concentrations of TEMPO_free_ (**A**) and serially diluted PY-LCNP-PEG/TEMPO (**B**) were used to measure cell variability using MTS assay. The 0.50× diluted PY-LCNP-PEG/TEMPO corresponds to a ~27 nM particle concentration containing 50 µM of conjugated TEMPO. Cells were incubated with TEMPO_free_ or PY-LCNP-PEG/TEMPO, washed, and cultured in growth medium for 72 h prior to the MTS assay. Average (*n* = 4 ± SEM) cell viability for different concentrations of TEMPO_free_ and PY-LCNP-PEG/TEMPO plotted as a percentage of control cells (not treated). Cell viability (>80%) of the comparable concentration (~50 µM) of TEMPO in a free or conjugated form are indicated by the arrows.

## Data Availability

Data is contained within the article.

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
