# Peer review of "Liquid Crystal Nanoparticle Conjugates for Scavenging Reactive Oxygen Species in Live Cells"

_pharmaceuticals, 2022, doi:10.3390/ph15050604_

Round 1
Reviewer 1 Report
Overall comment
Cellular uptake and some antioxidative effects of a TEMPO conjugate (145 nm in size with about 1.8k copies of TEMPO each) were reported. The manuscript provides adequate information suggesting that the work is novel and interesting with good conduct, however, major revisions should be considered to clarify some points which are listed.
Specific comments
- The title is suggested to be revised as it may not readily represent the work. The term ‘Nitroxide-Liquid Crystal Nanoparticle Conjugates’ and the phrase ‘Scavenging Reactive Oxygen Species in Live Cells’ are not clearly defined. If this title is to be used, nitroxide, liquid crystal and scavenging ROS in live cells must be clarified and fully explained in the context.
- The word ‘excellent’ (Line 326) might not be appropriate for the TEMPO conjugate. As it is, enhanced cellular uptake with some ROS reduction are shown. To state ‘excellent ROS reduction’ of the test material, ROS reduction should be at least to the same level as that found in the negative control.
- Lacks in information (methodology, result, discussion) to show and support these following findings:
- About 1.8k copies of TEMPO per particle.
- Colloidal stability.
Please add in the revision version.
- Was there any controlled condition during the evaporation of chloroform in the synthesis of nanoparticle conjugates, as chloroform should rapidly, not slowly, evaporated at 64 C.
- There are too many abbreviations used with or without necessities.
- Remove those abbreviations dealing with technical procedures, e.g. DIC, TRITC, FITC, etc., which are shown on the screens of the instrumental analysis equipment used by the researchers, but not the readers. These procedures should be written from the point of view of the readers (to know enough to understand the handling, obtaining and management of data).
- Selectively abbreviate only some terminologies and consistently use throughout.
- Liquid crystal nanoparticle, conjugate, nanoparticle, and colloid were used to define the developed material. Nitroxide-Liquid Crystal Nanoparticle Conjugate is used in the title but not elsewhere in the manuscript, why? Clear explanation to distinguish the differences among these terms followed by explaining the decision-making criteria used in selecting only one which best represents the developed material is essentially discussed.
- It is essential to clearly explain data management in the methodology section, e.g. fluorescent intensities, changes in intensities, etc. Readers should not guess by themselves.
- Please discuss the improvement of intracellular localization of TEMPO by the conjugates due to non- charged nature. (Lines 195-196)
- Terminology inconsistencies were numerous. Please carefully check and thoroughly revise.
Author Response
Overall comment
Cellular uptake and some antioxidative effects of a TEMPO conjugate (145 nm in size with about 1.8k copies of TEMPO each) were reported. The manuscript provides adequate information suggesting that the work is novel and interesting with good conduct, however, major revisions should be considered to clarify some points which are listed.
Author response: The authors would like to thank the reviewer for the comments/suggestions. We have addressed these carefully and made appropriate changes in the revised manuscript.
Specific comments
- The title is suggested to be revised as it may not readily represent the work. The term ‘Nitroxide-Liquid Crystal Nanoparticle Conjugates’ and the phrase ‘Scavenging Reactive Oxygen Species in Live Cells’ are not clearly defined. If this title is to be used, nitroxide, liquid crystal and scavenging ROS in live cells must be clarified and fully explained in the context.
Author response: Following the Reviewer’s excellent suggestion, we have changed the title to “Liquid Crystal Nanoparticle Conjugates for Scavenging Reactive Oxygen Species in Live Cells”. The terms TEMPO and scavenging reactive oxygen species are defined in the Introduction.
- The word ‘excellent’ (Line 326) might not be appropriate for the TEMPO conjugate. As it is, enhanced cellular uptake with some ROS reduction are shown. To state ‘excellent ROS reduction’ of the test material, ROS reduction should be at least to the same level as that found in the negative control.
Author response: The relevant sentence has been changed as follows-
“The conjugate showed significant cellular uptake, and enhanced efficacy in reducing ROS activity and lipid peroxidation when the cells were challenged with the ROS generating agents H2O2 and tBHP”
- Lacks in information (methodology, result, discussion) to show and support these following findings:
- About 1.8k copies of TEMPO per particle.
- Colloidal stability.
Author response:
- a) 1.8k copies of the TEMPO per particle:
The following sentence has been added to the revised manuscript.
“The extent of conjugation of TEMPO onto the PY-LCNP surface was determined spectroscopically using the TEMPO absorbance maximum at 426 nm (data not shown) and extinction coefficient (17.34 M-1 cm-1) [27]. TEMPO absorbance on the LCNP was deconvoluted by subtracting the absorbance of the LCNP.”
- b) Colloidal stability:
Although long-term colloidal stability of the PY-LCNP-PEG/TEMPO was not empirically determined, no visual aggregation or precipitation of the particle was observed upon weeks of storage at 4 degree. This is probably due to the conjugated PEG on the surface, which is well known for improved colloidal stability.
“excellent colloidal stability” has been removed from the abstract
The following sentence has been added to the relevant section of the revised manuscript (p. 3, line 152)–
“Fully characterized particles were stored at 4 °C and used within 3 weeks with no visually detectable precipitation or aggregation. This is likely attributable to the presence of PEG, which is well known for imparting colloidal and steric stability to NPs, on the surface of the particles.”.
Please add in the revision version.
- Was there any controlled condition during the evaporation of chloroform in the synthesis of nanoparticle conjugates, as chloroform should rapidly, not slowly, evaporated at 64 C.
Author response: This reaction was conducted under controlled N2 atmosphere. The relevant sentence has been changed to
“The mixture was then heated to 64 °C under N2 atmosphere to initiate the polymerization of both the cross-linking agent and surfactant as the chloroform evaporated, leaving PY-LCNP suspension stabilized by the surfactant” in the revised manuscript.
- There are too many abbreviations used with or without necessities.
- Remove those abbreviations dealing with technical procedures, e.g. DIC, TRITC, FITC, etc., which are shown on the screens of the instrumental analysis equipment used by the researchers, but not the readers. These procedures should be written from the point of view of the readers (to know enough to understand the handling, obtaining and management of data).
Author response: The abbreviations TRITC and FITC have been removed from the revised manuscript.
- Selectively abbreviate only some terminologies and consistently use throughout.
- Liquid crystal nanoparticle, conjugate, nanoparticle, and colloid were used to define the developed material. Nitroxide-Liquid Crystal Nanoparticle Conjugate is used in the title but not elsewhere in the manuscript, why? Clear explanation to distinguish the differences among these terms followed by explaining the decision-making criteria used in selecting only one which best represents the developed material is essentially discussed.
Author response: Inconsistent terminologies have been checked and fixed in the revised manuscript. The comment and comment 1 are relevant. The authors agree with changing the title of the manuscript to be more specific. Please see the response for comment 1.
- It is essential to clearly explain data management in the methodology section, e.g. fluorescent intensities, changes in intensities, etc. Readers should not guess by themselves.
Author response: Excellent comments. This issues have been checked in the revised manuscript.
- Please discuss the improvement of intracellular localization of TEMPO by the conjugates due to non- charged nature. (Lines 195-196)
Author response: Excellent suggestion. Upon EDC conjugation reaction TEMPO lost its primary amine functionality while decorated on the surface of the LCNP. The improved cellular uptake of TEMPO as PY-LCNP-PEG/TEMPO can be attributed to not only for the no-charge property but also for the overall modulation of its properties as PY-LCNP-PEG/TEMPO. The relevant sentence has been in the revised manuscript as follows-
“Thus, based on this prior literature, this suggests that our PY-LCNP-PEG/TEMPO would improve cellular uptake of TEMPO due to the overall modulation of its properties (e.g. non-charge nature due to the lack of primary amine after conjugation) attributed by the LCNP. ”
- Terminology inconsistencies were numerous. Please carefully check and thoroughly revise.
Author response: Inconsistent terminologies has been checked and fixed in the revised manuscript.
Reviewer 2 Report
Major remarks:
-The first part of "Results and Discussion" lines 100-107 concerns the introduction to manuscript, not results
-the image of the gel electrophoresis in Figure 1B is unclear. The electrophoresis should take longer to obtain a better separation image to evaluate the particles obtained
-line 421 - "manufacturer’s protocol." The catalog numbers of the reagents used are missing
Minor remarks:
-subsection numbering is missing
-"homeostasis In this study" a dot is missing between sentences
-line 63, "Yashitomi" should be "Yoshitomi"
-lines 74-75 - reference [20] should be moved after "Li et al"
-page 10, there are two dots at the end of the caption for Figure 6
- lines 337-338, these 2 compounds are the same or different, please correct
4-amino-2,2,6,6-tetramethylpiperidine-1-oxyl,
4-amino-2,2,6,6-tetramethylpiperidinyloxy (4-amino-TEMPO),
Author Response
Author response: The authors would like to thank the reviewer for the comments/suggestions. We have addressed these carefully and made appropriate changes in the revised manuscript.
Major remarks:
- The first part of "Results and Discussion" lines 100-107 concerns the introduction to manuscript, not results
Author response: Excellent suggestion. A subsection titled “Rationale for the synthesis of PY-LCNP-PEG/TEMPO” has been added for this section of the revised manuscript to help introduce the reader to the logic of the nanoparticle system design.
- The image of the gel electrophoresis in Figure 1B is unclear. The electrophoresis should take longer to obtain a better separation image to evaluate the particles obtained
Author response: As mentioned in the experimental section, this gel was ran for 30 min. Further running of the gel has found no meaningful change in the mobility (data not shown). This is sometimes seen in nanoparticle-conjugate systems. Fortunately, the DLS data and zeta potential data provided strong evidence for the successful conjugation of the tempo to the NP surface.
- line 421 - "manufacturer’s protocol." The catalog numbers of the reagents used are missing
Author response: Catalong numners have been added to the revised manuscript
Minor remarks:
- subsection numbering is missing
Author response: Thanks for pointing out this. Subsection numbering has been added to the revised manuscript.
- "homeostasis In this study" a dot is missing between sentences
Author response: The missing period has been added in the revised manuscript.
- line 63, "Yashitomi" should be "Yoshitomi"
Author response: This has been fixed in the revised manuscript.
- lines 74-75 - reference [20] should be moved after "Li et al"
Author response: Ref [20] has been interested in the end of the line.
- page 10, there are two dots at the end of the caption for Figure 6
Author response: The extra dot has been removed.
- lines 337-338, these 2 compounds are the same or different, please correct
Author response: This has been fixed in the revised manuscript.
- 4-amino-2,2,6,6-tetramethylpiperidine-1-oxyl, 4-amino-2,2,6,6-tetramethylpiperidinyloxy (4-amino-TEMPO)
Author response: We have made this nomenclature and abbreviations consistent throughout the revised manuscript.
Reviewer 3 Report
Overall, a very interesting study on surface-functionalized, fluorescent liquid crystalline systems. Some points to consider:
-What do the authors mean by "copies of TEMPO"? Can't they simple refer to TEMPO molecules?
-I am not sure if you should explain EDC the first time.
-How did the authors assess the PY-LCNP-PEG/TEMPO nanoparticle concentration, which also led them to the PY-LCNP/TEMPO ratio?
-What is the polydispersity of the developed systems? did the authors measure the pdi values through dls?
-What is the stability of these nanosystems, the LCNPs as well as the PEG/TEMPO conjugated ones? can they be preserved in liquid form or do they require further processing for preparing a stable formulation?
-Are there any images showing the morphology of these NPs?
-Please explain sem when it is first mentioned.
-The incubation period for all endocytosis experiments was only 30min? did the authors test longer incubation periods?
-Since the scavenging activity of the final nanosystem was evident, could it be the case that the PEG molecules do not cover and protect the nanoparticle surface? also, could PEG molecules be mostly absent from the surface? in other words, what is the utility and efficiency of PEG decoration in this study? are there any relative results, eg protein interactions?
Author Response
Overall, a very interesting study on surface-functionalized, fluorescent liquid crystalline systems. Some points to consider:
Author response: The authors would like to thank the reviewer for the comments/suggestions. We have addressed these carefully and made appropriate changes in the revised manuscript.
- What do the authors mean by "copies of TEMPO"? Can't they simple refer to TEMPO molecules?
Author response: “copies of TEMPO” has been replaced with “molecules of TEMPO” in the revised manuscript (line 25).
- I am not sure if you should explain EDC the first time.
Author response: EDC has been defined in the revised manuscript.
- How did the authors assess the PY-LCNP-PEG/TEMPO nanoparticle concentration, which also led them to the PY-LCNP/TEMPO ratio?
Author response: The particle concentration of PY-LCNP-PEG/TEMPO was determined by DLS. This was mentioned in the Results and Discussion, and Materials and Methods.
- What is the polydispersity of the developed systems? did the authors measure the pdi values through dls?
Author response: Polydispersity index for the particles was measured by DLS and this has been added to the revised manuscript (line 133).
- What is the stability of these nanosystems, the LCNPs as well as the PEG/TEMPO conjugated ones? can they be preserved in liquid form or do they require further processing for preparing a stable formulation?
Author response: In the course of use these particles remain stable at 4 °C. The following sentence has been added to the revised manuscript (line 152).
“Fully characterized particles were stored at 4 °C and used within 3 weeks with no visually detectable precipitation or aggregation.”
- Are there any images showing the morphology of these NPs?
Author response: Thank you for the comment. We feel that imaging particle morphology would not provide significant valuable data beyond that already provided by the DLS, zeta potential, and conjugation characterization.
- Please explain sem when it is first mentioned.
Author response: This change has been made in the revised manuscript (p.398).
- The incubation period for all endocytosis experiments was only 30min? did the authors test longer incubation periods?
Author response: For the internalization study, the particles were incubated up to 6 hours. Please see Figure 2 and the relevant results and discussion.
- Since the scavenging activity of the final nanosystem was evident, could it be the case that the PEG molecules do not cover and protect the nanoparticle surface? also, could PEG molecules be mostly absent from the surface? in other words, what is the utility and efficiency of PEG decoration in this study? are there any relative results, eg protein interactions?
Author response: Excellent questions. After PEG/TEMPO conjugation, the particles showed ~24 nm increase in their hydrodynamic diameter. TEMPO is a small molecule (MW 171 Da), while the PEG used in this reaction has MW 2000 Da. This significant increase in size has occured due to the conjugation of PEG. Additionally, PEG is well known for facilitating NP colloidal stability upon conjugation to the NP surface, which was mentioned in the manuscript. The following sentence has been added to the revised manuscript (line 152)-
“Fully characterized particles were stored at 4 °C and used within 3 weeks with no visually detectable precipitation or aggregation. This is probably due to the existence of PEG, which is well known for imparting colloidal and steric stability to NPs, on the surface of the particles”
Reviewer 4 Report
Overall, some corrections are needed in the manuscript:
Please list all chemicals in the materials section, possibly with CAS number as well.
Check the manuscript carefully for mistakes like 7 x104 cells (line 384); viabil-ity (line 309) etc.
Some abbreviations are not explained at their first use. (e.g. LCNP line 78). Check these.
Change in vivo and in vitro everywhere to italic type.
The following sentence is not needed in the materials section, please put it somewhere else: "This assay works based ... period." Possibly, to the introduction's last paragraph.
The last paragraph of the introduction should be revised, as it already mentions and explains experimental data. It should be focused on the short description of all methods used in the manuscript. For example, the first paragraph of the Results and Discussion part could be moved here.
Figure 1. A, B, and C are too compressed. Please enlarge these images. The same can be said about all other figures too.
Also, Figure 1. A. should be separated from the actual results, as it is visual description of the reaction.
Reformatting the y axis of Figure 3 B should be considered.
Figure 4. A and B should use the same symbols for μM.
Please further describe the statistical method used, as Tukey's test is a post-hoc test and no data is available about the normal distribution, etc.
The "Discussion" part of Results and Discussion must be further improved, to involve more comparison with literature results.
Author Response
Overall, some corrections are needed in the manuscript:
Author response: The authors would like to thank the reviewer for the comments/suggestions. We have addressed these carefully and made appropriate changes in the revised manuscript.
- Please list all chemicals in the materials section, possibly with CAS number as well.
Author response: Available CAS# for the chemicals have been added to the Materials section of the revised manuscript.
Check the manuscript carefully for mistakes like 7 x104 cells (line 384); viabil-ity (line 309) etc.
Author response: These have been fixed in the revised manuscript.
- Some abbreviations are not explained at their first use. (e.g. LCNP line 78). Check these.
Author response: Definition for LCNP has been added, and other abbreviations have been checked in the revised manuscript.
- Change in vivo and in vitro everywhere to italic type.
Author response: According to Pharmaceuticals (MDPI) formatting these do not required italicization.
The following sentence is not needed in the materials section, please put it somewhere else: "This assay works based ... period." Possibly, to the introduction's last paragraph.
Author response: This sentence has been removed from the experimental section of the revised manuscript.
The last paragraph of the introduction should be revised, as it already mentions and explains experimental data. It should be focused on the short description of all methods used in the manuscript. For example, the first paragraph of the Results and Discussion part could be moved here.
Author response: Excellent suggestion. More experiment method has been added to the last paragraph of the introduction.
The first paragraph of the “Results and Discussion” was indented to briefly describe the rationale of PY-LCNP-PEG/TEMPO synthesis. A subsection tittle “Rationale for the synthesis of PY-LCNP-PEG/TEMPO” has been added to the first paragraph of the as it briefly describes the rationale for the TEMPO conjugation to the LCNP.
- Figure 1. A, B, and C are too compressed. Please enlarge these images. The same can be said about all other figures too.
Author response: These images were enlarged to their maximum size to fit the journal page formatting and template.
- Also, Figure 1. A. should be separated from the actual results, as it is visual description of the reaction.
Author response: Thank you for the suggestion; we feel that the figure is effective in its current format.
- Reformatting the y axis of Figure 3 B should be considered.
Author response: The y axis was formatted with two segments break to enhance the visual separation among the data sets.
- Figure 4. A and B should use the same symbols for μM.
Author response: This inconsistency has been fixed in the revised manuscript.
- Please further describe the statistical method used, as Tukey's test is a post-hoc test and no data is available about the normal distribution, etc.
Author response: Thank you for pointing this out. For multiple comparisons, Bonferroni’s post hoc test was applied. The following correction has been made to the relevant section (line 455)-
“For multiple comparisons, Bonferroni’s post hoc test was applied. All average values were given ± standard error of mean. The acceptable probability for significance was p < 0.05”
- The "Discussion" part of Results and Discussion must be further improved, to involve more comparison with literature results.
Author response: An additional discussion point has been added to the revised manuscript (line 319).
..”Additionally, NPs based ROS scavenging agents, such as nanoceria, have been reported to inhibit lipid peroxidation at the mitochondrial level and deter the progression of Parkinson’s disease [31]. One of the implications of our PY-LCNP-PEG/TEMPO can be the inhibition of lipid peroxidation in subcellular locations, such as mitochondria, when designed to be specifically delivered”
Round 2
Reviewer 1 Report
Accept.
Reviewer 2 Report
Can be published in the present form
Reviewer 4 Report
I would like to thank the authors for addressing all listed points. I am happy, that the overall quality of the manuscript is increased and it equals the quality and importance of the results as well.